# How Does Long-COVID Impact Prognosis and the Long-Term Sequelae?

**DOI:** 10.3390/v15051173

**Published:** 2023-05-15

**Authors:** Carolina Baroni, Jorge Potito, María Eugenia Perticone, Paola Orausclio, Carlos Marcelo Luna

**Affiliations:** 1Department of Medicine, Pulmonary Diseases Division, Hospital de Clínicas, University of Buenos Aires, Buenos Aires C1120 AAF, Argentina; 2Department of Radiology, Centro Rossi, Buenos Aires C1035 ABC, Argentina

**Keywords:** long-COVID, sequelae, long-term, prognosis, SARS-CoV-2

## Abstract

Context: We reviewed what has been studied and published during the last 3 years about the consequences, mainly respiratory, cardiac, digestive, and neurological/psychiatric (organic and functional), in patients with COVID-19 of prolonged course. Objective: To conduct a narrative review synthesizing current clinical evidence of abnormalities of signs, symptoms, and complementary studies in COVID-19 patients who presented a prolonged and complicated course. Methods: A review of the literature focused on the involvement of the main organic functions mentioned, based almost exclusively on the systematic search of publications written in English available on PubMed/MEDLINE. Results: Long-term respiratory, cardiac, digestive, and neurological/psychiatric dysfunction are present in a significant number of patients. Lung involvement is the most common; cardiovascular involvement may happen with or without symptoms or clinical abnormalities; gastrointestinal compromise includes the loss of appetite, nausea, gastroesophageal reflux, diarrhea, etc.; and neurological/psychiatric compromise can produce a wide variety of signs and symptoms, either organic or functional. Vaccination is not associated with the emergence of long-COVID, but it may happen in vaccinated people. Conclusions: The severity of illness increases the risk of long-COVID. Pulmonary sequelae, cardiomyopathy, the detection of ribonucleic acid in the gastrointestinal tract, and headaches and cognitive impairment may become refractory in severely ill COVID-19 patients.

## 1. Introduction

At the beginning of 2020, the world became aware of the outbreak of a respiratory infection, caused by a new coronavirus, that started in the city of Wuhan, China [1]. No one could imagine at that time that COVID-19, on the one hand, would make a huge number of inhabitants sick (undoubtedly many more than what the records show), and on the other hand, it would change our vision regarding much of the knowledge that characterized the pandemics that had devastated our planet in the past. At the onset of the pandemic, there was great concern about the presence of potential complications, especially those due to lung involvement, particularly acute respiratory distress syndrome (ARDS) and respiratory failure, in a similar way to the way it had happened during the previous decades with the coronaviruses responsible for SARS and MERS [2]. In severely affected individuals, the destruction of lung tissue and the consequent deterioration in blood gas exchange leading to arterial hypoxemia ended up compromising their basic vital functions [2,3,4].

Although it will be near impossible to know how many people have suffered from COVID-19 during the last 3 years, it appears to be easier to estimate the number of people who suffered from a long course of illness since the more severe and prolonged manifestations make the presence of the virus more ostensible. After the World Health Organization (WHO) recognized the persistence of symptoms for more than two months (Long-COVID) as an emerging health problem, it was assigned an ICD code for emergency use [5]. In September 2022, the WHO estimated that at least 17 million people in the WHO European Region experienced long-COVID in the first two years of the pandemic and millions may have to live with it for years to come [6]. Two years after the debut of this overwhelming disease, we have seen many of its characteristics and have learned to handle many of the challenges it presents to us, but we are still learning about the prognosis and long-term consequences of this disease [7].

At this point, it is critically important to review the accumulated evidence about the long-term consequences of COVID-19 in terms of persistent systemic tissue damage and clinical repercussion in the medium and long term to address the management of these patients. Thus, it might be possible to explore the usefulness of traditional medicine, vaccines [8], and newer therapeutic approaches and associated pathologies with long-COVID [9].

We wrote this review about the prognosis and the long-term associated pathologies with long-COVID more than 3 years after the start of the pandemic, updating the available evidence about the long-term consequences that COVID-19 presents us. Severe acute respiratory syndrome coronavirus 2 (SARS-CoV-2) presents us with a complex range of manifestations that can affect multiple organ systems and probably leave impressive long-term sequelae [10]. While the scientific community seeks to characterize the long-term consequences of SARS-CoV-2 infection, this review tries to shed light on this issue by drawing on what has been published in recent years. These efforts that try to examine the prognosis and the long-term consequences of COVID-19 may be useful to better understand this clinical condition and thus be able to mitigate its consequences.

## 2. Definition of Long-COVID

Early after the onset of the COVID-19 pandemic, it was frequently observed that after the acute phase, many patients, even some who had suffered mild symptoms, presented persistent systemic signs and symptoms of varying severity, constituting what has been called long-COVID that could last from several weeks to months [11].

The National Research Action Plan on Long-COVID and the Services and Support for the Longer-term Impacts of COVID-19 in the United States proposed the following interim definition: “Long-COVID is broadly defined as signs, symptoms, and conditions that continue or develop after initial SARS-CoV-2 infection. The signs, symptoms, and conditions are present four weeks or more after the initial phase of infection; may be multisystemic; and may present with a relapsing–remitting pattern and progression or worsening over time, with the possibility of severe and life-threatening events even months or years after infection. Long-COVID is not one condition, it represents many potentially overlapping entities, likely with different biological causes and different sets of risk factors and outcomes” [12].

The long-term consequences of COVID-19 began to be studied in mid-2020 when cases of this disease were affecting most of the world. Huang et al. conducted one of the first studies on a cohort of 1733 patients discharged from a hospital in Wuhan, China, over12 months in 2020 [13,14]. They found that at 6 and 12 months after acute infection 68% and 49% (*p* < 0.0001) of COVID-19 survivors presented at least one sequelae symptom, respectively. The results showed that 26wenty-six and 30% (*p* = 0.014) showed sleep difficulties and 23% and 26% (*p* = 0.015) experienced anxiety or depression, respectively. They also found that at one year, compared with men, women had an odds ratio of 1.43 (95% CI 1.04–1.96) for fatigue or muscle weakness, an odds ratio of 2.00 (1.48–2.69) for anxiety or depression, and an odds ratio of 2.97 (1.50–5.88) for diffusion impairment. The most significant change observed in this study, the improvement of symptomatic sequelae in 6 months, contrasts with the study by Wu et al., who found that a subgroup of hospitalized patients who did not require mechanical ventilation had persistent physiological and radiographic changes after 1 year [15].

## 3. Clinical Presentation

Patients who have suffered from COVID-19 may develop in severe cases persistent manifestations of pulmonary, cardiovascular, neurological, gastrointestinal, hematological, renal, and endocrine compromise, among others, for a prolonged duration and with variable intensity after acute infection [12]. Interdisciplinary monitoring with holistic management that considers nutrition, physical therapy, psychological management, meditation, and mindfulness in addition to medication allows for the early detection of post-acute COVID-19 sequelae symptoms and the prevention of long-term systemic damage [16]. This review describes the main aspects of the clinical presentation of long-COVID in four more commonly affected systems: cardiovascular, respiratory, gastrointestinal, and neurological and psychiatric domains, with a clinical presentation different from that found in acute COVID-19 (Table 1 and Figure 1).

## 4. Pulmonary Domain

COVID-19 was described in a cluster of 41 hospitalized patients in the city of Wuhan, China, as severe pneumonia characterized by bilateral pulmonary infiltrates in 40 (98%) cases and requiring invasive or non-invasive mechanical ventilation in 14 (34%) cases [1]. Lingering after-effects including “inflamed lungs” were described 120 years ago following the “Russian influenza”. Persistent pulmonary complications such as impaired exercise capacity, decreased diffusion lung capacity, and interstitial pulmonary infiltrates have been described in SARS-CoV-1 survivors [17].

The impact on the respiratory system represents one of the most common and potentially serious situations and is related to the exaggerated immune response triggered by the infection [18]. Cytokine storm and its consequences are responsible for the tissue damage of the lung parenchyma [19]. The most common pulmonary manifestations include symptoms of fatigue and dyspnea, a cough, and the presence of ground-glass opacities and fibrotic changes on CT scans [13,20]. Diseases associated with the development of chronic symptoms include bacterial pneumonia [21], respiratory failure [22], thromboembolic pulmonary disease [23], pulmonary vascular damage [24,25], and post-viral pulmonary fibrosis [18]. Persistent respiratory symptoms beyond 1 month are a frequent finding; dyspnea is a frequently observed complaint, especially in patients who have been hospitalized and particularly in those who require intensive care. Some of these patients develop chronic respiratory failure. Still, around 10% of patients who did not require hospitalization may have dyspnea after having suffered from acute COVID-19 [26,27]. As expected, the outcome is much worse in COVID-19 patients requiring a tracheostomy for prolonged invasive mechanical ventilation, in which the weaning process is successful in less than half of them [28].

Studies have shown abnormalities in functional tests at 3 months in non-critical patients in up to 25% of cases, frequently with decreased diffusing capacity for carbon monoxide (DLCO) [20]. The functional defect, if present, tends to improve in most patients. A persistent cough is a less common symptom but can last for a long time. Some patients who have had COVID-19 may present with persistent dyspnea and other symptoms in the absence of evident pulmonary or cardiovascular involvement presenting with a condition called dysfunctional breathing including hyperventilation syndrome [29]. This disorder may require a cardiopulmonary exercise test to make the correct diagnosis.

With regards to radiological sequelae, the gold standard study is high-resolution computed tomography (HRCT) which allows for a high-quality analysis of the parenchyma of reversible and irreversible parenchymal lesions.

Myall et al. reported, in an observational original research study on 35/837 survivors of COVID-19 with persistent non-improving symptoms and significant interstitial lung disease with physiological and functional impairment 6 weeks after discharge, that 30 patients who had received steroids showed a mean relative increase in transfer factor following treatment of 31.6% (standard deviation [SD] ± 27.6, *p* < 0.001) and forced vital capacity of 9.6% (SD ± 13.0, *p* = 0.014), with significant symptomatic and radiological improvement [30].

Pulmonary fibrosis is the archetype of irreversible post-COVID injury. It is characterized histologically by the destruction of the normal lung structure partially replaced by a reconstruction with an extracellular collagen matrix [31]. Interstitial abnormalities include reticulation of the parenchyma, with the presence of traction bronchiectasis, and even honeycombing [32]. The frequency of this presentation varies according to the authors and at what point in the evolution of COVID-19 the HRCT was performed. These lesions are more common and complex in seriously ill patients, particularly the most critical ones complicated by bacterial infection and/or ARDS [33], (Figure 2).

In some patients with severe, unresolved COVID-19-associated ARDS lung, transplantation is considered the only option for survival. The procedure has been performed successfully in 12 patients at 6 high-volume transplant centers in the United States, Italy, Austria, and India with good early outcomes [31,34].

## 5. Cardiovascular Domain

Post-acute cardiovascular sequelae of SARS-CoV-2 infection include new, returning, or persistent health conditions that can be present not only in individuals 4 or more weeks after a documented or suspected SARS-CoV-2 infection, but also after mRNA vaccination against SARS-CoV-2 [35,36]. The risk of developing cardiovascular sequelae depends on many factors related to the severity of the initial infection and the patient’s baseline condition. Xie et al. estimated the risk of cardiovascular disease after an acute infection as 52% for stroke and 72% for heart failure [37].

Cardiovascular symptoms, such as dyspnea, chest pain, and palpitations, with or without specific laboratory and imaging findings have been described [10]. Patients also might present with hypertension, tachycardia disproportionate with mild effort, and the finding of hypoxia in SpO_2_ during physical examination. There are specific cardiovascular entities that can be present in this population (myocarditis, pericarditis, and myocardial damage). In addition, abnormal complementary studies have been found in asymptomatic patients [38,39,40].

The American College of Cardiology separates cardiovascular sequelae into two well-defined conditions: cardiovascular diseases and cardiovascular symptoms. The first refers to established cardiovascular entities that can be diagnosed during the early post-acute or chronic periods of COVID-19 and the second refers to symptoms that appear after the acute phase of the disease and are not fully explained by the cardiovascular complementary studies that fail to show abnormalities [41].

The distinction between these entities may depend on the study algorithm and the chosen image method. Moreover, some methods are more sensitive and can detect incipient abnormalities, and those findings might be present in asymptomatic individuals. In addition, there is no clear evidence that indicates that the identified abnormalities are related to SARS-CoV-2 infection or pre-existing conditions or are related to the current symptoms [42,43].

Long-COVID-related diseases such as myocarditis, pericarditis, myocardial ischemia, non-ischemic cardiomyopathy, thromboembolic disease with or without cardiac compromise (pulmonary hypertension and right ventricular dysfunction), and arrhythmias, are the most commonly described entities. It is challenging to determine whether these conditions are new or pre-existing, sometimes because of the timing of the presentation or because the patient might delay seeking medical attention [44].

On the other hand, the long-COVID cardiovascular conditions include a heterogeneous variety of symptoms in patients without evidence of myocardial injury. The most frequently reported symptoms are tachycardia, exercise intolerance, dyspnea, and chest pain. Some other accompanying symptoms include fatigue, cognitive complaints, sleep disturbance, and post-exertional malaise. Postural orthostatic tachycardia syndrome (POTS) and myalgic encephalomyelitis/chronic fatigue syndrome are two entities that are described after viral infections and can be present in patients with a recent history of COVID-19 [45].

Data from different studies indicate that 10–30% of patients may experience long-term symptoms after an acute mild SARS-CoV-2 infection [46]. Cardiovascular symptoms are commonly reported in published series as they are present in up to 86–89% of patients. They are described as fluctuant or relapsing and can persist as long as 7 months after the initial infection [40,47,48]. It is important to note that many of the studies are based on self-reported symptoms from patients who have experienced a recent SARS-CoV-2 infection. In those reports, cardiovascular symptoms are as high as 85% of all persistent symptoms [49]. Almost 20% of the patients reported chest pain at 60 days of follow-up, and at 6 months, palpitations and chest pain were reported in 9 and 5% of all patients, respectively [13,14,38].

There is not a demonstrated correlation between symptoms and the diagnosis of cardiovascular disease. Two studies found that only a minority of patients were found to have a specific cardiovascular diagnosis despite a complete evaluation. [50,51]

Patients also report exercise intolerance, fatigue, or malaise and a 30 beats per minute increase in heart rate with the standing position, which can be related to neuro-hormonal changes. Deconditioning (especially related to bedrest) can be a trigger and a maintenance mechanism for these symptoms, especially in those who have experienced moderate to severe infection. The role of deconditioning in patients with mild infection is not completely understood [49].

There is evidence that supports that chest pain may be induced by coronary vasospasm without evidence of coronary artery disease, myocarditis, or pericarditis. This is explained by the endothelial injury secondary to an exaggerated immune response. Unexplained dyspnea in this group of patients may be related to deconditioning and poor cardiovascular status. This symptom is more commonly reported in women, but there is no clear evidence that suggests an association with other variables. Up to 88% of patients may show abnormalities when a cardio-pulmonary exercise test is performed, but there is no specific pattern [52,53,54,55].

The complete physiopathology of cardiovascular involvement is not completely understood. Studies suggest that different mechanisms are involved, such as increased cardiometabolic demand, reduced cardiac reserve, neurohormonal dysregulation (the renin–angiotensin–aldosterone system), myocardial inflammation, fibrosis, immune activation, the persistence of viral load, endothelial dysfunction, impaired exercise metabolism, and cardiac deconditioning [31,56,57].

A multidisciplinary approach is suggested. Initially, a complete clinical evaluation should be performed, which might include basic laboratory determinations, electrocardiograms, chest images, Holter monitoring, and pulmonary function tests. The patient should be referred to a cardiologist to perform a more complete evaluation if there are abnormal results [58,59].

Additionally, it is reasonable to wait until 4 weeks after acute disease for a diagnostic evaluation to avoid unnecessary studies [41].

## 6. Gastrointestinal Domain

At the gastrointestinal level, SARS-CoV-2 interacts with and infects gastrointestinal cells by contacting the angiotensin-converting enzyme 2 receptor found in intestinal epithelial cells, with the small intestine being the site of the greatest localization of these receptors [38]. This interaction of SARS-CoV-2 with the receptor at the gastrointestinal level translates into potential acute or chronic gastrointestinal manifestations.

The presence of the RNA of SARS-CoV-2 in feces reveals the interaction of the virus at the gastrointestinal level, with diarrhea being the main manifestation. Digestive symptoms are common in patients with long-COVID [60]. These gastrointestinal clinical manifestations can last over time as part of the gastrointestinal picture of the prolonged syndrome caused by COVID-19; on the other hand, the virus has been detected in stool samples even after resolution of respiratory symptoms [39,61]. The presence of viral RNA in stools suggests the presence of infectious viral particles secreted by the infected gastrointestinal cells. After viral infection, the virus triggers several reactions to produce specific RNA and proteins that are then released into the gastrointestinal tract [62]. The presence of viral RNA in feces tells us that the virus continues to infect the gastrointestinal tract and can cause prolonged clinical manifestations over time that are included in what is known as the gastrointestinal condition of prolonged COVID-19 syndrome.

Huang et al. described that 76% of patients who were hospitalized after a documented infection of SARS-CoV-2 presented at least one symptom that persisted for at least 6 months after the disease, with gastrointestinal symptoms being one of them [12].

Gastrointestinal symptoms that can last over time or even appear months after infection and that were not present at the time of infection or prior to it are a loss of appetite, nausea, gastroesophageal reflux, and diarrhea [63]; less frequent are the gastrointestinal sequelae developed in long-COVID syndrome, such as abdominal distension, abdominal pain, and bloody stools.

Gastrointestinal sequelae including a loss of appetite, nausea, acid reflux, and diarrhea are clinical manifestations that may present upon discharge of the patient who required hospitalization for COVID-19; these persistent gastrointestinal symptoms that are part of the gastrointestinal condition present in prolonged COVID-19 syndrome have implications for the daily life of patients as well as for the resources allocated to the proper management of these patients.

Patients who have suffered severe pneumonia with decreased blood oxygen saturation values have been shown to be more prone to developing gastrointestinal sequelae. It has been suggested that this might be due to the presence of multi-organ dysfunction syndrome that can develop in severe COVID-19 pneumonia in the context of septic shock [64,65].

It should also be noted that gastrointestinal symptoms such as nausea, vomiting, and diarrhea present during hospitalization for SARS-CoV-2 infection can lead to malnutrition, which has been linked to increased mortality in patients with COVID-19 [66,67]. Nutrition plays a fundamental role not only during hospitalization but also in reducing gastrointestinal sequelae.

## 7. Neurological and Psychiatric Domain

Several persistent neurological findings have been described in patients who have suffered from COVID-19, with them still being present months after the onset of the infection. These symptoms include fatigue, headache, myalgia, weakness, vertigo, dysgeusia, anosmia, etc. [13,65]; such symptoms have become the hallmark of long-COVID syndrome. In addition to the longstanding neurological and sensory consequences described in patients with COVID-19 more than 4 weeks after the onset of the disease, post-traumatic stress disorder and psychiatric disorders such as anxiety, depression, insomnia, and memory loss might also be present in several of these patients [12]. The persistence of these symptoms, particularly fatigue and sleep disturbances, may have an impact on quality of life, productivity, and mental health [68,69]. The need for social support during hospitalization with a more robust approach to managing uncertainty regarding health status and family concerns is important. The severity of the disease, ICU admission, and certain drugs used during the acute infection may increase the risk of neurological sequelae in the post-acute COVID-19 period [37].

Few studies have explored the cognitive, psychological, and quality of life consequences of long-COVID in hospitalized and non-hospitalized patients. Fatigue is a commonly found symptom in convalescent COVID-19 patients, with it persisting for several weeks or months after the end of the disease [69].

Interestingly Premraj et al. in a meta-analysis looking at the prevalence of neurological and neuropsychiatric symptoms reported ≥3 months post onset of COVID-19 observed that the prevalence of neurological, cognitive, and psychological symptoms, such as anxiety and depression, was found to be lower among hospitalized patients than among non-hospitalized patients (31% and 27% vs. 16% and 12%, respectively) [70]. Anosmia, dysgeusia, and headache are common during the acute phase of COVID-19 but tend to resolve in less than 6 weeks in 60 to 70% of patients [71].

The COVID-19 pandemic has threatened global mental health, both through neuropsychiatric sequelae following infection and indirectly through consequential disruptive social changes. Recent public opinion survey data has shown that the COVID-19 pandemic has affected the mental health of society [72].

The COVID-19 pandemic and the different health policies implemented generated a great social and economic impact with detrimental effects on quality of life due to its physical and mental consequences on the population. The consequent reported increase in long-term mental problems secondary to COVID-19 reveals both acute and post-acute neuropsychiatric sequelae. Indirect effects, such as disruptive social changes deteriorating mental health, such as direct effects, including acute and long-lasting neuropsychiatric sequelae, that could happen as part of long-COVID syndrome and can affect multiple organs including the brain, have been previously described in infections caused by other pathogenic coronaviruses [73].

Different studies have demonstrated that the severity of the disease during the acute phase (admission to the ICU and the need for mechanical ventilation) is associated with the presence or persistence of symptoms, deterioration of health-related quality of life, and organic and functional respiratory abnormalities in the post-acute setting of COVID-19 [10,74,75,76].

The causative role of systemic inflammatory damage in the brain is supported by the fact that none of the cerebrospinal fluid samples investigated in the study by Helms et al. found evidence of the direct presence of SARS-CoV-2 [70,77].

## 8. Treatment and Rehabilitation Impact on Prognosis and Long-Term Sequelae

Long-COVID is being recognized as a systemic disease with a diverse spectrum of multi-organ manifestations. This made it necessary to address aspects of the multidisciplinary management of convalescent patients who had suffered from COVID-19, focusing on those who are at high risk of presenting persistent symptoms [11]. Those people at high risk of long-COVID should be screened among those who had presented a severe acute disease and include those admitted to an intensive care unit (ICU), those who had required oxygen therapy or mechanical ventilation, and those who presented with a persistent cough, dyspnea, fatigue, persistent or worsening radiographic abnormalities in the post-acute setting, or worsening of quality of life or pulmonary function tests [12,13,78] (Table 1).

More severely ill patients with acute COVID-19 are at higher risk of developing a long-COVID clinical course and complications with functional impairment and persistent radiological abnormalities. Pulmonary thrombosis has been described in up to 25% of patients with COVID-19, particularly in those who are critically ill [79]. On the other hand, the severity of the endothelial injury and widespread thrombosis with microangiopathy seen on lung autopsy is greater than that seen in ARDS caused by influenza and may predispose to bacterial colonization and subsequent severe COVID-19 ventilator-associated pneumonia [80,81].

Hematological sequelae: The rate of venous thromboembolism in the post-acute COVID-19 setting being <5% COVID-19-associated coagulopathy is consistent with a hyperinflammatory and hypercoagulable state [82,83]. The risk of thrombotic complications in the post-acute COVID-19 phase could be linked to the presence of a hyperinflammatory state and how long this persists. A D-dimer level greater than two times the upper limit of normal, associated with comorbidities and immobility, increases the risk of post-acute thrombosis; however, the risk/benefit equation should be considered before prescribing anticoagulation [84]. Direct oral anticoagulants and low-molecular-weight heparin are preferred anticoagulation agents over vitamin K antagonists due to the lack of a need to frequently monitor therapeutic levels as well as the lower risk of drug–drug interaction [55].

Cardiovascular sequelae: A higher incidence of cardiomyopathy has been observed during the COVID-19 pandemic; however, the rate of rehospitalization or mortality does not appear to have increased for this reason [85]. The magnetic resonance imaging study of these patients suggests the presence of persistent myocardial inflammation in most of them for more than 2 months after the diagnosis of COVID-19, which must be confirmed by other studies [78]. Autonomic dissociation responsible for tachycardia, sometimes associated with orthostatic hypotension, has been described.

Gastrointestinal sequelae: There are few sequelae derived from gastrointestinal involvement in COVID-19 survivors [68]. There is prolonged viral fecal shedding confirmed by ribonucleic acid detected for nearly 1 month after the onset of symptoms. Viral detection persists for more than 10 days after negativizing airway samples [86].

There is limited data about the long-term consequences on the gastrointestinal tract, but there are studies running looking at the possible consequences including the development of irritable bowel syndrome [87].

Neuropsychiatric sequelae: Neurological complications such as headaches should receive standard treatment; however, when such headaches become refractory the patient should be referred to a specialist for evaluation [88]. Long-term cognitive impairment is a widely known clinical condition in patients who have been hospitalized in an ICU. In these patients, a reasonable time after the onset of symptoms without improvement should be considered for further neuropsychological evaluation. In these cases, it is necessary to use standard screening tools to identify the presence of disorders such as anxiety or depression, sleep disturbances, fatigue, or dysautonomia [20].

## 9. What it the Effect of Vaccines on the Incidence and Severity of Long-COVID?

The first vaccines available for the prevention of COVID-19 have shown during clinical trials a level of efficacy between 80.2% for adenovirus-based vaccines and 94.6% for mRNA-based vaccines [89]. Given the incidence of asymptomatic disease, which has been suggested to be as high as 30% [90], both before and after vaccination, their true efficacy is very difficult to know with certainty. On the other hand, emerging variants after the development of the different vaccines, particularly widely spread in the case of Omicron, have been associated with immune escape [91,92].

Major complications, including multi-organ failure which emerged after vaccination, have also been reported; however, Rabaan et al. in a systematic review found that there have been no studies to establish a correlation between vaccination and multi-organ failure [93].

To date, few useful studies have been carried out to reliably determine the impact of immunization with vaccines on the incidence and severity of long-COVID. Peghin et al. performed a prospective study on 479 adults diagnosed with COVID-19 during the first wave in 2020 and cared for and followed up at an academic hospital at 6 and 12 months [94]. The patients were interviewed one year after acute infection and post-COVID-19 syndrome was observed in 47.2% of them in this non-vaccinated population of COVID-19 patients. There were no significant differences in the worsening of post-COVID symptoms (22.7% vs. 15.8%; *p* = 0.209) among the vaccinated and unvaccinated patients. Additionally, the presence of serological assays (receptor-binding domain (RBD) SARS-CoV-2 IgG) to distinguish between the response to vaccination showed a significant association with post-COVID-19 syndrome, and median IgG titers were significantly higher in long-term patients than in patients without symptoms, with the study concluding that SARS-CoV-2 but not vaccination is associated with the emergence of long-COVID [82].

On the other hand, Gao, et al. performed a study aimed at exploring the relationship between vaccination and long-COVID, reviewing relevant studies in the real world, and performed a meta-analysis to explore the aforementioned relationship. They found that the vaccinated group had a 29% lower risk of developing long-COVID compared with the unvaccinated group. Vaccination was effective against long-COVID in patients either vaccinated before or after SARS-CoV-2 infection. For long-COVID symptoms, vaccination reduced the risk of cognitive dysfunction/symptoms, kidney diseases/problems, myalgia, and sleeping disorders [95].

Marra et al. in another systematic literature review and meta-analysis on the effectiveness of COVID-19 vaccination against post-COVID conditions (including long-COVID) found that the pooled diagnostic odds ratios for post–COVID-19 conditions among individuals vaccinated with at least 1 dose showed estimated vaccine effectiveness of 29.2% (35.3% among those who received the COVID-19 vaccine before having COVID-19 and 27.4% among those who received it after having COVID-19) [81]. They concluded that during the study period, vaccination both before and after having COVID-19 significantly decreased post-COVID-19 conditions for the circulating variants even though the vaccine’s effectiveness was low [96].

## 10. Conclusions

During these long 3 years of the COVID-19 pandemic, an unexpected path has been marked out for us: most of the patients are treated for their illness on an outpatient basis; 15%, more frequently in the elderly population, suffer from a complicated disease that requires hospitalization; and about 5% develop a critical illness with respiratory failure and mortality close to 50%. A significant number of patients, especially those more vulnerable due to their advanced age or suffering from comorbidities, and those with a more serious disease, develop a complicated clinical course with long convalescence, the persistence of different symptoms, and sequelae that have been called long-COVID. In these cases, there is usually multi-systemic compromise; the most frequently affected systems are usually the respiratory, cardiovascular, gastrointestinal, and neurological/psychiatric systems. After this long period of time, the world has adjusted to a new normality; the herd immunity produced after the contagion brought by the pandemic and the widely extended vaccination could not prevent repeated waves of reappearance of the problem with different variants including Omicron, which, although in a context of a high percentage of the vaccinated population, showed certain limits to the effectiveness of immunization. It also revealed a potential role in attenuating symptoms of acute infection, convalescence, and sequelae.

The pathogenesis of long-COVID is clearly multifactorial and difficult to unravel. It is still not clear what role the persistence of the virus has in different organs, its reactivation, and the long-term immune response. Even when the acute impact of COVID-19 fades, the prognosis and sequelae of this disease will continue to require our attention for a long time, particularly in those who developed long-COVID. If so, this review will be a summary of what is known and a starting point that can better guide the knowledge of the associated pathophysiological mechanisms for better prevention and treatment for addressing its management.

## Figures and Tables

**Figure 1 viruses-15-01173-f001:**
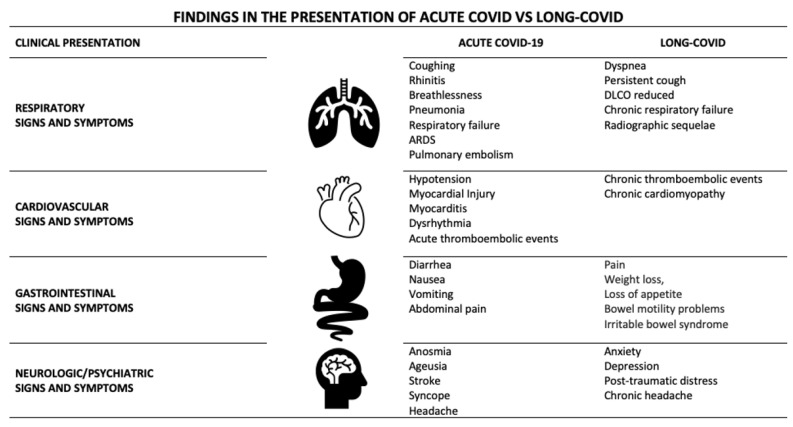
Findings in the presentation of acute COVID vs. long-COVID.

**Figure 2 viruses-15-01173-f002:**
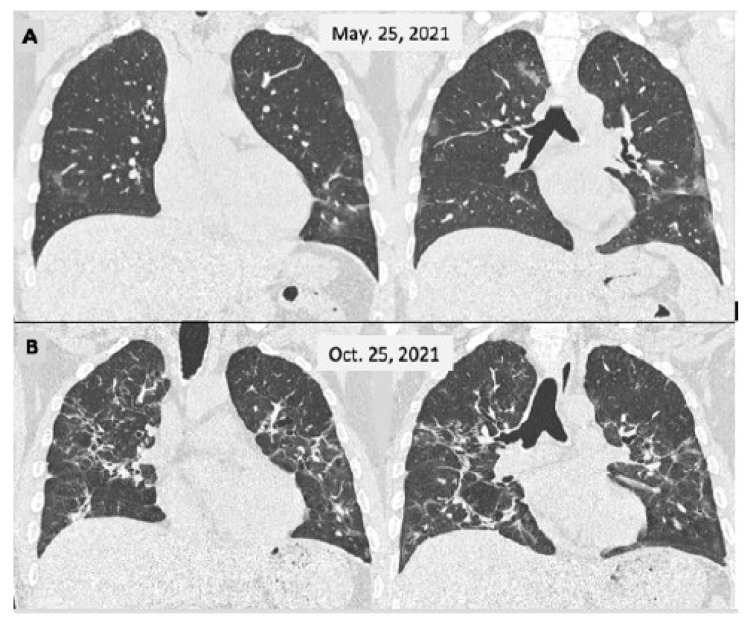
Male, 46 y-o, admitted on 25 March 2021 with a fever and breathlessness. Their PCR test was positive for SARS-CoV-2. The CT scan (**A**) showed a few small and faint bilateral peripheral opacities, mainly in the lower lobes (moderate COVID-19); the patient was cared for as an outpatient. He persisted to have a fever for 2 weeks and displayed slow improvement over the next 2 months with a persistent cough. A second CT scan (**B**) was performed 5 months later showing more extensive lesions with sequelae appearance.

**Table 1 viruses-15-01173-t001:** Differences in the presentation of acute COVID vs long COVID.

	Acute-COVID	Long-COVID
Respiratory Signs and Symptoms	Coughing Rhinitis Breathlessness Pneumonia Respiratory failure ARDS Pulmonary embolism	Dyspnea Persistent cough DLCO reduced Chronic respiratory failure Radiographic sequelae
Cardiovascular Signs and Symptoms	Hypotension Myocardial Injury Myocarditis Dysrhythmia Acute thromboembolic events	Chronic thromboembolic events Chronic cardiomyopathy
Gastrointestinal Signs and Symptoms	Diarrhea Nausea Vomiting Abdominal pain	Pain Weight loss, Loss of appetite Bowel motility problems Irritable bowel syndrome
Neurologic/Psychiatric Signs and Symptoms	Anosmia Ageusia Stroke Syncope Headache	Anxiety Depression Post-traumatic distress Chronic headache

## Data Availability

No new data was created.

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
