# Peer review of "How Does Long-COVID Impact Prognosis and the Long-Term Sequelae?"

_viruses, 2023, doi:10.3390/v15051173_

Round 1

Reviewer 1 Report (Previous Reviewer 1)

This is an interesting and useful review. This is a well-structured and well-illustrated account. I recommend this paper “How does Long-COVID impact the Prognosis and the Long-Term Sequelae?” to be published in the journal. The following comment should be addressed:

Page 2, “Two years after the debut of this over-whelming disease, we have known many of its characteristics and have learned to handle many of the challenges it presented to us, but we are still learning about the prognosis and long-term consequences of this disease.” Please refer to the following papers of the significant progress (antibodies, small molecule agents [Front. Immunol. 2022, 13, 1015355], traditional medicines, vaccines [JAMA. 2021, 325, 1318-1320]) which will make it easier for the reader to find relevant sources in mitigating the threat of COVID-19.  Please refer to the following papers:

Author Response

REVIEWER 1

This is an interesting and useful review. This is a well-structured and well-illustrated account. I recommend this paper “How does Long-COVID impact the Prognosis and the Long-Term Sequelae?” to be published in the journal. The following comment should be addressed

  1. Page 2, “Two years after the debut of this over-whelming disease, we have known many of its characteristics and have learned to handle many of the challenges it presented to us, but we are still learning about the prognosis and long-term consequences of this disease.” Please refer to the following papers of the significant progress (antibodies, small molecule agents [Front. Immunol. 2022, 13, 1015355], traditional medicines, vaccines [JAMA. 2021, 325, 1318-1320]) which will make it easier for the reader to find relevant sources in mitigating the threat of COVID-19.  

Response: Thank you for your comment. Indeed, as the reviewer suggests, beyond the ignorance that accompanies doctors in the management of this new disease, it is necessary at some point to mention the tools that we have at hand to try to mitigate the long-term effects of COVID-19. We have added this concept and some references including those suggested by Reviewer 1.

Reviewer 2 Report (Previous Reviewer 2)

Dear Authors,

The manuscript is a narrative review, but I have serious concerns about the originality of the theme and approach. Unfortunately, narrative and systematic reviews are already present in the literature;

even reviews on the most appropriate semantics linked to

the definition of long-covid (https://www.mdpi.com/1660-4601/18/5/2621 ),

also insights on the sexual perspective (https://doi.org/10.1016/j.sxmr.2021.11.001 )

and dermatological (https://doi.org/10.1016/S1473-3099(20)30986-5 )

What I can suggest is to change the methodological approach. Maybe there are the conditions for a critical review or a mapping review?

Regarding the manuscript: again I would increase the data in support of the discursive nature.

"Vaccination is not associated with the emergence of Long-COVID, but it may happen in vaccinated people". I don't understand this feed-forward syllogism. We can just say: Vaccination is not associated with the emergence of Long-COVID.

Minor editing of English language required

Author Response

REVIEWER 2

  1. The manuscript is a comprehensive narrative review, but I have serious concerns about the originality of the theme and approach. Unfortunately, narrative and systematic reviews are already present in the literature; even reviews on the most appropriate semantics linked to the definition of long-covid (https://www.mdpi.com/1660-4601/18/5/2621), also insights on the sexual perspective (https://doi.org/10.1016 /j.sxmr.2021.11.001) and dermatological (https://doi.org/10.1016/S1473-3099(20)30986-5). The manuscript is a narrative review, but I have serious concerns about the originality of the theme and approach. Unfortunately, narrative and systematic reviews are already present in the literature.

Response: Thank you for this important comment about our narrative review. We agree that the topic of this review article has already been the subject of a significant number of manuscripts in the last 3 years that have enriched the knowledge of this mode of presentation and evolution of COVID-19. When VIRUSES invited us to write it, along with the initial invitation, the topic and the title were included, and at that moment it seemed to us that it was something interesting that deserved to be done with a vision of what had been learned during the last 3 years. 

  1. What I can suggest is to change the methodological approach. Maybe there are the conditions for a critical review or a mapping review?

Response: When we were invited to write a review on Long-COVID-12 we were not required to use a special review style and that is why we naturally chose the narrative style. We agree that a different methodological approach could help us reach different conclusions but doing so would force us to replan the review from another perspective, with another methodology, which is a challenge that we might consider addressing but at the same time discarding the manuscript that much time and effort took us. In our response to the following points, we explain how we try to improve the profile of this review by increasing the number of our sources of information which are basically the most recent publications on this topic.

  1. Regarding the manuscript: again, I would increase the data in support of the discursive nature.

Response: We have discarded the old abstract and replaced it with a structured abstract. We also recognize that in some of the portions of the manuscript it is necessary to increase the data support of the discursive nature, so following this comment we increased our data supporting trying to improve the discursive nature of our review.

  1. Vaccination is not associated with the emergence of Long-COVID, but it may happen in vaccinated people". I don't understand this feed-forward syllogism. We can just say: Vaccination is not associated with the emergence of Long-COVID.

Response: we removed the sentence about the association of SARS-CoV-2 variants with immune escape since there are no results or conclusions in our review. At the end of the paragraph dedicated to this topic we added the paragraph “vaccination is not associated with the emergence of Long-COVID, as suggested by Reviewer 2.

  1. Comments on the Quality of English Language: Minor editing of English language required

Response: We reviewed the manuscript with a US-born English professor familiar with writing and translating English texts and medical terminology and edited it under his guidance.

Round 2

Reviewer 2 Report (Previous Reviewer 2)

-

This manuscript is a resubmission of an earlier submission. The following is a list of the peer review reports and author responses from that submission.

Round 1

Reviewer 1 Report

The manuscript needs some work before it is ready for the show.

1: As the COVID-19, an outbreak caused by the Severe Acute Respiratory Syndrome Coronavirus 2 (SARS-CoV-2), is still spreading and this study may have a certain amount of readership. In general, this clinical experience is sound, but take “COVID-19 infection” (lines 169, 279) for example, the authors should carefully reviewed the manuscript to unify as “SARS-CoV-2 infection”. This concern should be taken into consideration.

2: line 58, “SARS-CoV-2” should corrected to “Severe Acute Respiratory Syndrome Coronavirus 2 (SARS-CoV-2)”.

3: line 243, “COVID-19 virus” should corrected to “SARS-CoV-2”.

4: line 245, “SARS-CoV-2 virus” should corrected to “SARS-CoV-2”.

5: To be complete, the authors should add plenty of references. For example, in lines 58-59, “The SARS-CoV-2 presents us with a complex range of manifestations that can affect multiple organ systems and probably leave impressive long-term sequelae.(DOI: 10.3389/fpubh.2022.908757.)”; in lines 97-100, “Interdisciplinary monitoring with holistic management that considers nutrition, physical therapy, psychological management, meditation, and mindfulness in addition to medication allow for the early detection of post-acute COVID-19 sequelae symptoms and prevent long-term systemic damage.(DOI: 10.1038/s41591-021-01283-z) ”; in lines 148-150, “Pulmonary fibrosis is the archetype of irreversible post-COVID injury. From the point of view characterized histologically by the presence of destruction of the normal lung structure partially replaced by a reconstruction with an extracellular collagen matrix.(DOI: 10.26355/eurrev_202103_25438)”.

6: The “Conclusion” is weak. Authors should thoroughly improve it because it's the final impression.

Reviewer 2 Report

Dear Authors,

The manuscript is a comprehensive narrative review, but I have serious concerns about the originality of the theme and approach. Unfortunately, narrative and systematic reviews are already present in the literature; even reviews on the most appropriate semantics linked to the definition of long-covid (https://www.mdpi.com/1660-4601/18/5/2621 ), also insights on the sexual perspective (https://doi.org/10.1016 /j.sxmr.2021.11.001  ) and dermatological (https://doi.org/10.1016/S1473-3099(20)30986-5  )

What I can suggest is to change the methodological approach. Maybe there are the conditions for a critical review or a mapping review?

Regarding the manuscript: in addition to suggesting a structure to the abstract, I would increase the data in support of the discursive nature.

21 "SARS-CoV-2 variants are associated with immune escape". I suggest removing it, they are not results or conclusions of your review

22 I disagree. “Low level of evidence (grade III, case-controls, cohort studies) suggests that vaccination before SARS-CoV-2 infection could reduce the risk of subsequent long-COVID. “(https://doi.org/10.1016/j.eclinm.2022.101624 )

Abbreviations are not described in the manuscript

55 “We accepted the challenging invitation to write this review” remove, please

Table 1, could be modified as a figure, I suggest to deepen it.. among other things, signs and symptoms do not imply: DLCO reduced, Chronic Respiratory Failure, furthermore the Radiographic Sequelae are contemplated in the acute.